# Clinical Profile and Management of Patient Patients with Ischemic Heart Disease and/or Peripheral Artery Disease in Clinical Practice: The APALUSA Study

**DOI:** 10.3390/jcm11123554

**Published:** 2022-06-20

**Authors:** Vivencio Barrios, Carlos Escobar, Carmen Suarez, Xavier Garcia-Moll, Francisco Lozano

**Affiliations:** 1Cardiology Department, University Hospital Ramon y Cajal, 28034 Madrid, Spain; 2Cardiology Department, University Hospital La Paz, 28046 Madrid, Spain; carlos.escobar@salud.madrid.org; 3Internal Medicine Department, University Hospital la Princesa, 28006 Madrid, Spain; csuarezf@salud.madrid.org; 4Cardiology Department, University Hospital de la Santa Creu i Sant Pau, 08025 Barcelona, Spain; xgmoll@gmail.com; 5Vascular Surgery Department, University Hospital Clínico, 37007 Salamanca, Spain; lozano@usal.es

**Keywords:** antiplatelet therapy, blood pressure, cardiovascular, diabetes, ischemic heart disease, LDL cholesterol, peripheral artery disease

## Abstract

This study was aimed to ascertain the clinical profile and management of patients with ischemic heart disease (IHD) and/or peripheral artery disease (PAD). In this observational and cross-sectional study developed in 80 hospitals throughout Spain, consecutive adults with stable IHD and/or PAD were included. A total of 1089 patients were analyzed, of whom 65.3% had only IHD, 17.8% PAD and 16.9% both. A total of 80.6% were taking only one antiplatelet agent, and 18.2% were on dual antiplatelet therapy (mainly aspirin/clopidogrel). Almost all patients were taking ≥1 lipid lowering drug, mainly moderate-to-high intensity statins. IHD patients took ezetimibe more commonly than PAD (43.9% vs. 12.9%; *p* < 0.001). There were more patients with IHD that achieved blood pressure targets compared to PAD (<140/90 mmHg: 67.9% vs. 43.0%; *p* < 0.001; <130/80 mmHg: 34.1% vs. 15.7%; *p* < 0.001), LDL-cholesterol (<70 mg/dL: 53.1% vs. 41.5%; *p* = 0.033; <55 mg/dL: 26.5% vs. 16.0%; *p* = 0.025), and diabetes (HbA1c < 7%, with SGLT2i/GLP1-RA: 21.7% vs. 8.8%; *p* = 0.032). Modifications of antihypertensive agents and lipid-lowering therapy were performed in 69.0% and 82.3% of patients, respectively, without significant differences between groups. The use of SGLT2i/GLP1-RA was low. In conclusion, cardiovascular risk factors control remains poor among patients with IHD, PAD, or both. A higher use of combined therapy is warranted.

## 1. Introduction

Atherosclerotic cardiovascular disease is the leading cause of morbidity and mortality in Western countries [1]. Atherosclerosis, which is comprised of ischemic heart disease (IHD) and peripheral artery disease (PAD), among others, is a progressive chronic condition with a high risk of recurrent cardiovascular events, in the same or other vascular beds [2,3,4]. For example, it has been estimated that around 50 to 75% of patients with a history of myocardial infarction will have a new cardiovascular event within 1 to 3 years after the event [1,2,5]. Similarly, PAD is associated with a markedly increased risk of coronary and cerebrovascular disease [6]. 

To actually reduce cardiovascular burden in patients with atherosclerotic cardiovascular disease, it is necessary to understand that atherosclerosis is a multifactorial condition that is influenced not only by the negative impact of cardiovascular risk factors (i.e., hypertension, dyslipidemia, diabetes, smoking) but also by a pro-aggregating state due to endothelial dysfunction [7,8]. Therefore, only through a comprehensive approach in patients with atherosclerotic cardiovascular disease can the risk of recurrent events in secondary prevention patients be reduced [9]. Thus, in a study performed among patients with IHD, the combination of aspirin, statins and angiotensin-converting enzyme inhibitors (ACEi) reduced all-cause mortality by 71% (vs. 47% with statins alone and 61% with the combination of statins plus aspirin) [10].

Although many studies have been developed in patients with IHD or PAD to analyze cardiovascular risk factors control and antithrombotic therapy, the fact is that the majority of these studies have been focused only on one condition, one cardiovascular risk factor control or on only antithrombotic therapy [11,12,13,14,15,16]. As a result, it is necessary to develop well-designed studies that analyze the comprehensive management of patients with atherosclerotic cardiovascular disease, either IHD, PAD or both. A better knowledge of the contemporary clinical profile and therapeutic approach of this population would improve the management of these patients in clinical practice. 

The APALUSA (estudio observacional para evaluar el manejo clínico y la **A**decuación de las estrategias terapéuticas utilizadas en **P**acientes con enfermedad **A**terosclerótica estab**L**e y el **U**so de los tratamiento**S** recomendados de **A**cuerdo a las guías clínicas [observational study to evaluate the clinical management and adequacy of therapeutic strategies used in patients with stable atherosclerotic cardiovascular disease and the use of treatments recommended by clinical guidelines]) study was aimed to ascertain the clinical profile and the therapeutic management of patients with IHD and/or PAD, according to the European Society of Cardiology (ESC) clinical guidelines. In addition, cardiovascular risk factors’ control and adherence to ESC guidelines were also investigated. 

## 2. Methods

This was an observational and cross-sectional study developed in the departments of cardiology (*n* = 45), internal medicine (*n* = 20) and vascular surgery (*n* = 15) of 80 hospitals throughout Spain. Consecutive adult patients with stable IHD and/or PAD and that provided written informed consent were included. Inclusion and exclusion criteria are summarized in Table 1. The study was approved by the clinical research ethics committee of the University Hospital Clinic of Barcelona, Spain, on July 2019 and was endorsed by the local Institutional Review Boards of each participating center. The first patient was recruited on 20 November 2019 and the last patient on 4 February 2021. The recruitment of patients was affected by COVID-19 pandemic.

The study contained only one visit that coincided with any of the routine follow-up visits of the patients according to clinical practice. No specific diagnostic procedure or therapeutic approach was performed for enrollment in the study. Only available data from the clinical history or during the interview with the physician were collected. All data were recorded using an electronic case report form specially designed for this study.

At baseline, biodemographic parameters, data from physical examination (blood pressure, heart rate, weight, height, body mass index), cardiovascular risk factors (dyslipidemia, hypertension, diabetes, sedentarism, smoking habits, family history of cardiovascular disease), cardiac disease (prior myocardial infarction, heart failure) and blood analysis obtained within 6 months before being enrolled (HbA1c, renal function, complete lipid profile) were recorded. In addition, treatments at baseline were also collected: antiplatelet therapy (single or dual antiplatelet, type of agent and duration of treatment), lipid lowering therapy (statins -type, dose and duration-, ezetimibe, fibrates, proprotein convertase subtilisin/kexane 9 (PCSK9) inhibitors), renin angiotensin system inhibitors (ACEi, angiotensin receptor blockers (ARB), aldosterone antagonists, sacubitril/valsartan), antianginal drugs (beta blockers, calcium channel blockers, nitrates, ivabradine, ranolazine, cilostazol), and revascularization procedures (coronary, PAD or carotid revascularization, either percutaneous or surgical). Data were analyzed according to the presence of IHD and/or PAD (IHD or PAD, IHD and PAD, IHD alone, PAD alone).

The proportion of patients that achieved cardiovascular risk factor targets was determined. For blood pressure [17], <140/90, <130/80 and <120/70 mmHg were considered. For lipids, LDL cholesterol targets of <70 mg/dL and <55 mg/dL were considered, as 2019 dyslipidemia ESC guidelines had been recently published at the time of the study [18,19]. An additional target of <50 mg/dL was also considered. For diabetes, HbA1c levels (<7% and <6.5%) and the use of sodium-glucose cotransporter-2 inhibitors (SGLT2i) or glucagon-like peptide 1 receptor agonists (GLP1-RA) were determined [20]. Data were analyzed according to the presence of IHD and/or PAD (IHD or PAD, IHD and PAD, IHD alone, PAD alone).

Adherence to 2017 ESC guidelines regarding the prolongation of antiplatelet therapy was evaluated [21,22]. Among IHD patients [21], good adherence was considered in case of prolongation of dual antiplatelet therapy beyond one year after the initiation of the treatment in those patients with a score ≥ 2 in DAPT scale (age, cigarette smoking, diabetes, myocardial infarction at presentation, prior percutaneous coronary intervention or myocardial infarction, type and diameter of stent, heart failure or left ventricular ejection fraction < 30%, and vein graft stent) without an event [23]. Among patients with PAD [22], good adherence was considered in the case of prolongation of aspirin or clopidogrel beyond one year after the initiation of the treatment, if tolerated. With regard to cardiovascular prevention drugs, adherence to 2013 ESC guidelines was analyzed [24]: (1) the proportion of patients with IHD treated with low-dose aspirin (or clopidogrel if not tolerated); (2) the proportion of patients treated with statins; (3) the proportion of patients with heart failure or left ventricular dysfunction treated with ACEi or ARB; (4) proportion of patients with hypertension or diabetes treated with ACEi or ARB. 

Finally, the actions recommended/taken during the visit by the physician according to clinical practice were also collected. These actions included nonpharmacological recommendations (quit smoking, regular physical activity, healthy diet, weight loss), pharmacological recommendations (modification of antihypertensive drugs, lipid lowering therapy, antidiabetic drugs), and other interventions (i.e., cardiac rehabilitation, flu and *Pneumococcal* vaccines) [9,25]. Data were analyzed according to the presence of IHD and/or PAD (IHD or PAD, IHD and PAD, IHD alone, PAD alone).

## 3. Statistical Analysis

Qualitative variables were expressed as absolute and relative frequencies and quantitative variables as measures of central tendency and dispersion (mean and standard deviation). The baseline clinical profile and treatments, the proportion for patients achieving cardiovascular risk factors control and actions taken during the visit were compared between those patients with IHD alone and PAD alone. Categorical variables were compared using the chi-square test or the Fisher exact test, when appropriate, and means with the *t*-test. Statistical significance was set at 0.05. The data were analyzed using the statistical package SPSS (v18.0 or superior).

## 4. Results

A total of 1112 patients were initially enrolled. After excluding 23 (2.1%) due to different reasons (not meeting with inclusion/exclusion criteria, lack of data, discrepancies with data), 1089 (97.9%) patients were finally analyzed, of whom 711 (65.3%) had only ischemic heart disease (IHD), 194 (17.8%) peripheral artery disease (PAD) and 184 (16.9%), both conditions.

Baseline clinical characteristics are shown in Table 2. Overall, mean age was 68.9 ± 10.1 years, 78.8% had dyslipidemia, 72.3% hypertension, 57.9% prior myocardial infarction, and 38.7% diabetes. Mean blood pressure was 132 ± 17/76 ± 11 mmHg. With regard to biochemical parameters, mean LDL cholesterol was 73 ± 29 mg/dL and HbA1c 6.7 ± 1.6%. Compared with patients with IHD, patients with PAD were more frequently smokers, had a more sedentary lifestyle, had diabetes and higher values of systolic blood pressure, heart rate, HbA1c, total cholesterol, LDL cholesterol and triglycerides, but a lower proportion of patients with family history of cardiovascular disease, prior myocardial infarction and heart failure. 

Treatments at baseline are presented in Table 3. Almost all patients were taking antiplatelets, without significant differences between groups. The majority of patients (80.6%) were taking only one antiplatelet agent, and 18.2% were on dual antiplatelet therapy. Patients with IHD alone were more commonly on dual antiplatelet therapy than PAD patients (20.0% vs. 6.8%; *p* < 0.001). The majority of patients on single antiplatelet therapy (87.4%) were taking aspirin, and the most common combination in those on dual antiplatelet therapy was aspirin plus clopidogrel (58.1%). Among patients with IHD alone on dual antiplatelet therapy, 50.7% were taking aspirin plus clopidogrel, and 46.4% aspirin plus ticagrelor. Almost all patients were taking at least one lipid-lowering drug, mainly statins. Moderate to high intensity atorvastatin and rosuvastatin were the most common statins prescribed. Patients with IHD alone were taking ezetimibe more commonly than PAD patients (43.9% vs. 12.9%; *p* < 0.001). Less than 3% of patients were taking PCSK9 inhibitors. There were more patients with IHD taking renin angiotensin system inhibitors compared to PAD patients (75.1% vs. 64.4%; *p* = 0.004). As expected, antianginal drugs and coronary revascularization were more common among patients with IHD than in patients with PAD. 

More patients with IHD achieved blood pressure targets compared to PAD patients (<140/90 mmHg: 67.9% vs. 43.0%; *p* < 0.001; <130/80 mmHg: 34.1% vs. 15.7%; *p* < 0.001), as well as LDL cholesterol targets (<70 mg/dL: 53.1% vs. 41.5%; *p* = 0.033; <55 mg/dL: 26.5% vs. 16.0%; *p* = 0.025), and diabetes (HbA1c < 7%, with SGLT2i or GLP1-RA: 21.7% vs. 8.8%; *p* = 0.032) (Table 4, Figure 1).

Adherence to 2013 ESC guidelines recommendations was high with regard to antiplatelet therapy (77.8%) and the use of statins (97.7%), but low regarding the use of ACEi or ARB (Table 5). Good adherence to 2017 ESC guidelines regarding the prolongation of antiplatelet therapy was achieved in 73.0% of patients with IHD (mean DAPT score 3.0 ± 1.0) and in 97.6% of patients with PAD.

Table 6 shows the actions recommended/taken by the physicians during the visit. Overall, in only 56.0% of patients smoking, abstention was recommended. Increasing physical activity was recommended in only 59.1% of patients, and losing weight was recommended in 32.2% of cases. Recommendations about diet and physical activity were more commonly performed among IHD patients. Modifications of antihypertensive agents and lipid lowering therapy were performed in 69.0% and 82.3% of patients, respectively, without significant differences between groups. The use of SGLT2i and GLP1-RA was low in all groups, but higher among IHD patients with diabetes, regarding SGLT2i. Although cardiac rehabilitation was recommended in more patients with IHD, these figures were low.

## 5. Discussion

Our study showed that patients with either IHD, PAD or both still have a markedly poor control of cardiovascular risk factors and that adherence to guidelines is suboptimal. Despite that, and the fact that these patients exhibit many comorbidities, actions taken by physicians regarding nonpharmacological and pharmacological treatment are clearly insufficient.

In our study, a wide sample of patients with IHD and/or PAD were included. Mean age was around 69 years, and the proportion of patients with cardiovascular risk factors and concomitant conditions was high. This is in line with previous studies that have also shown that secondary prevention patients have a high-risk clinical profile [11]. The clinical profile was even worse in those patients with IHD (vs. PAD patients), particularly when both conditions coexisted. A recent analysis of the EUROASPIRE V study showed that among patients with IHD, 6.4% of the patients already had a confirmed diagnosis of PAD and another 6.3% suspected PAD [14]. In our study, 17% of patients had IHD and PAD, but it should be considered that the inclusion criteria were different in both studies. As in our study, in EUROASPIRE V registry, those patients with both conditions had a worse risk factor profile [14], indicating that in these patients a more aggressive approach is required to reduce the risk of incident events.

With regard to antithrombotic treatment, the majority of patients were taking only one antiplatelet agent, mainly aspirin. When dual antiplatelet therapy was prescribed, the most common combination was aspirin plus clopidogrel, followed by aspirin plus ticagrelor. Of note, the 2017 ESC guidelines’ recommendations regarding the prolongation of antiplatelet therapy were not followed in 27% of patients with IHD. Similar numbers have been reported by other authors [26]. Antiplatelet therapy represents the cornerstone to reduce the risk of new ischemic events in secondary prevention patients [9,21,22,24]. In fact, compared with placebo, antiplatelet therapy reduces the risk of recurrent MACE in patients with IHD and PAD [27,28]. However, despite single antiplatelet strategy, residual risk for cardiovascular events remains unacceptably high. In fact, around 5 to 10% of patients with atherosclerotic cardiovascular disease treated with one antiplatelet agent have recurrent cardiovascular events every year [29]. In the PEGASUS trial, among patients who had had a myocardial infarction one to three years earlier, compared with aspirin, dual antiplatelet therapy with ticagrelor and aspirin significantly reduced the risk of MACE, but not cardiovascular death, and increased the risk of major bleeding [30]. More recently, the COMPASS trial showed in patients with stable atherosclerotic vascular disease that compared with aspirin alone, the combination of rivaroxaban 2.5 mg twice daily plus aspirin translated into a lower risk of MACE and cardiovascular death, but with an increased risk of major bleeding [31]. In addition, although the inclusion criteria were different, residual risk was lower in the COMPASS trial than in the PEGASUS trial. This is not surprising since both, increased platelet activity and coagulation cascade have been involved in the etiopathogenesis of atherothrombosis [32]. All these data suggest that dual antithrombotic approach of aspirin plus either a second antiplatelet agent or low-dose rivaroxaban should be considered in more patients with atherosclerotic vascular disease to reduce cardiovascular burden. In fact, data from the REACH registry showed that more than a half of CAD or PAD patients could be candidates for a COMPASS strategy [33]. However, the risk of recurrent MACE and major bleeding should be considered individually [21].

Less than two thirds of patients attained the blood pressure goal of <140/90 mmHg and 30% the target of <130/80 mmHg. Of note, these figures were even worse among PAD patients. With regard to treatments, approximately three quarters of patients were taking renin angiotensin system inhibitors, two thirds beta blockers and one quarter calcium channel blockers. Despite the poor blood pressure control rates, physicians did not modify antihypertensive treatment in around 30% of patients. Although these numbers are better than those previously reported [34], they remain high, and more efforts are required to improve blood pressure control in this high-risk population. 

Traditionally, lipids control has been poor in secondary prevention patients [35]. Although more effective treatments are available, and there is more evidence about the benefits of achieving LDL cholesterol targets in this population [19,36], in our study, a target of LDL cholesterol < 70 mg/dL was achieved in nearly half of patients and <55 mg/dL in approximately one quarter of patients. Once again, these figures were even worse among PAD patients. These poor results were directly related with a low use of ezetimibe and PCSK9 inhibitors. Fortunately, in the majority of patients, lipid-lowering therapy was modified in order to obtain better LDL cholesterol control. In contrast with 2019 ESC guidelines that considered a step-by-step approach, delaying the achievement of LDL cholesterol goals, current recommendations promote the early use of lipid-lowering combined therapy to attain recommended targets [19,36].

The proportion of patients with diabetes that attained the double aim of HbA1c and the use of antidiabetic cardiovascular protective drugs (i.e., SGLT2i and GLP1-RA) was low in all groups, particularly in PAD patients. In fact, our study showed that the modification of antidiabetic treatment in this population was insufficient. Despite the benefits that these drugs have demonstrated in clinical trials, their use in clinical practice remains markedly low [37,38].

In summary, despite the high-risk profile, the proportion of patients achieving cardiovascular risk factor control targets was dramatically low. As there is reimbursement for the great majority of cardiovascular drugs in Spain, the main reasons for poor cardiovascular risk factors control may include insufficient medical education, the wrong perception that patients were adequately controlled that led to an inadequate intensification of treatment and the underestimation of cardiovascular risk [36,39,40]. Among patients who developed myocardial infarction from the Atherosclerosis Risk in Communities (ARIC) study, achieving recommended targets, including smoking, adiposity, physical activity, diet, total cholesterol, blood pressure, and fasting glucose, was associated with better prognosis in later life [41]. Unfortunately, our numbers were unacceptably low, mainly due to an insufficient optimization of both nonpharmacologic and pharmacologic therapies. For example, recommendations performed by physicians about changes in lifestyle were low. This has also been observed in the EUROASPIRE V study [42]. These data strongly suggest that although in all patients, lifestyle should be evaluated and healthy changes recommended, the fact is that mainly due to lack of time, advice about this matter are not frequently given [9]. 

Of note, our study showed that cardiovascular risk factors control was markedly poorer in patients with PAD than with IHD. This was clearly related to the lower use of protective cardiovascular drugs, and also fewer recommendations about physical activity, healthy diet and weight loss. Previous studies have also shown poor cardiovascular risk factor control among PAD population, mainly related to a low use of cardiovascular agents [43,44,45]. However, these patients benefit not only from smoking cessation but also from a healthy lifestyle and a higher use of antihypertensive drugs and lipid-lowering agents [1,46].

This study has some limitations. Because this was an observational and cross-sectional study, there was no control group and no direct conclusions can be inferred. However, this is the best design to make a clear picture of the clinical profile and management of patients in clinical practice. On the other hand, patients could be somewhat different according to the criteria they entered in the study, as definitions of IHD and PAD included a wide range of patients. Furthermore, patients were consecutively included as patients met with the inclusion/exclusion criteria. Unfortunately, the proportion of patients that did not present at this visit was not recorded, and this could have overestimated the proportion of patients that reached the targets. In addition, other vascular beds, such as atherosclerotic renal artery stenosis, were not analyzed. However, the high number of patients included, as well as the robustness of the data, make the results representative of patients with atherosclerotic cardiovascular disease. Finally, because we focused on Spain in this study, the generalizability of the results can only be applied to countries with a similar clinical profile and healthcare system.

In conclusion, patients with either IHD, PAD, or both exhibit a high-risk clinical profile. Cardiovascular risk factors control rates remain poor in this population, and more efforts are required to improve these figures, mainly through the promotion of healthy lifestyle changes and a higher use of combined therapy. In addition, dual antithrombotic therapy is underused in clinical practice. The comprehensive management of these patients is the best way to reduce the cardiovascular burden and the risk of recurrent MACE. 

## Figures and Tables

**Figure 1 jcm-11-03554-f001:**
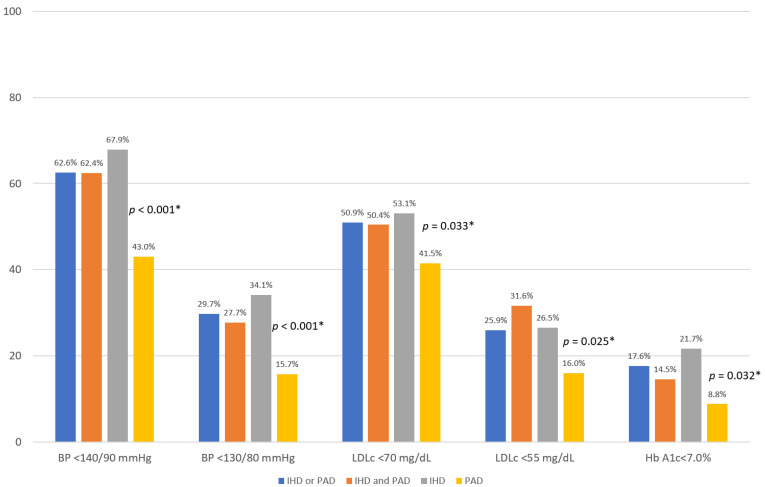
Proportion of patients achieving cardiovascular risk factor targets. BP: blood pressure; LDLc: LDL cholesterol; HbA1c: glycated hemoglobin; IHD: ischemic heart disease; PAD: peripheral artery disease. * *p*-value IHD vs. PAD.

**Table 1 jcm-11-03554-t001:** Inclusion/exclusion criteria and cardiovascular risk factors targets.

Inclusion Criteria	Exclusion Criteria
Consecutive adult patients with stable IHD and/or PAD: ∘ Stable IHD was defined as follows: ▪ (1) The history of previous myocardial infarction, or ▪ (2) Diagnosis of multivessel coronary disease with two major epicardial arteries having >50% stenosis, with symptoms or history of stable or unstable angina, or ▪ (3) Multivessel coronary disease revascularization, either percutaneous or surgical, within 1 to 10 years before inclusion. ▪ (4) In addition, patients should be ≥65 years or <65 years with revascularization of ≥2 major epicardial arteries, or with ≥2 additional risk factors (i.e., smoking, diabetes, eGFR < 60 mL/min/1.73 m^2^, heart failure or nonlacunar ischemic stroke). ∘ PAD was defined as: ▪ (1) Any revascularization of lower extremities, or ▪ (2) Previous vascular amputation of any part of lower extremities, or ▪ (3) Arterial stenosis ≥50%, or ▪ (4) History of intermittent claudication, or ▪ (5) An ankle–brachial index <0.9, or ▪ (6) Patients with carotid disease, including previous revascularization or arterial stenosis ≥50%. Patients should provide written informed consent	Patients hospitalized at inclusion Patients with a history of atrial fibrillation or other indication for anticoagulation
Cardiovascular risk factors targets	
Blood pressure [17]	<140/90 mmHg <130/80 mmHg <120/70 mmHg
Lipids [18,19]	LDL cholesterol < 70 mg/dL LDL cholesterol < 55 mg/dL
Diabetes [20]	HbA1c < 7% and treatment with SGLT2i or GLP1-RA HbA1c < 6.5% and treatment with SGLT2i or GLP1-RA

eGFR: estimated glomerular filtration rate; SGLT2i: sodium-glucose cotransporter-2 inhibitors; GLP1-RA: glucagon-like peptide 1 receptor agonists.

**Table 2 jcm-11-03554-t002:** Baseline clinical characteristics.

	IHD or PAD (*n* = 1089; 100%)	IHD and PAD (*n* = 184; 16.9%)	IHD (*n* = 711; 65.3%)	PAD (*n* = 194; 17.8%)	P_IHD_ vs. _PAD_
**Biodemographic data and physical examination**					
Age, years	68.9 ± 10.1	70.9 ± 9.3	68.6 ± 10.4	68.1 ± 9.8	0.561
SBP, mmHg	132 ± 17	133 ± 17	129 ± 15	141 ± 19	<0.001
DBP, mmHg	76 ± 11	75 ± 11	75 ± 11	77 ± 13	0.057
HR, bpm	68.0 ± 11.7	68.6 ± 11.1	66.0 ± 10.4	76.0 ± 13.9	<0.001
Weight, kg	79.7 ± 14.4	78.5 ± 11.5	80.0 ± 15.0	79.9 ± 15.0	0.975
BMI, kg/m^2^	27.9 ± 4.1	28.1 ± 3.8	27.8 ± 4.1	28.3 ± 4.1	0.067
**Cardiovascular risk factors**					
Dyslipidemia	78.8%	83.2%	77.2%	80.4%	0.380
Hypertension	72.3%	85.9%	68.4%	73.7%	0.160
Diabetes	38.7%	58.7%	31.5%	45.9%	<0.001
Sedentarism	33.1%	37.0%	29.7%	42.3%	0.001
Smoking					<0.001
No smoker	28.6%	20.6%	34.5%	15.5%	
Smoker	16.1%	15.0%	11.9%	31.4%	
Former smoker	55.4%	64.4%	53.6%	53.1%	
Family history of CVD	15.4%	10.9%	18.1%	9.8%	0.006
**Cardiac disease**					
Prior myocardial infarction	57.9%	64.1%	70.2%	0	<0.001
Heart failure	4.1%	7.6%	4.4%	0	0.001
**Biochemical parameters**					
HbA1c, %	6.7 ± 1.6	7.2 ± 1.8	6.4 ± 1.3	7.3 ± 2.2	<0.001
eGFR (mL/min/1.73 m^2^)	72.6 ± 20.3	69.4 ± 21.2	73.9 ± 19.9	71.2 ± 20.4	0.374
Total cholesterol, mg/dL	144 ± 35	144 ± 36	140 ± 33	162 ± 40	<0.001
HDL cholesterol, mg/dL	47 ± 13	45 ± 14	47 ± 12	50 ± 16	0.271
LDL cholesterol, mg/dL	73 ± 29	71 ± 28	69 ± 26	91 ± 40	<0.001
Triglycerides, mg/dL	130 ± 65	146 ± 65	121 ± 60	149 ± 75	<0.001

BMI: body mass index; CVD: cardiovascular disease; DBP: diastolic blood pressure; eGFR: estimated glomerular filtration rate; HR: heart rate; IHD: ischemic heart disease; PAD: peripheral artery disease; SBP: systolic blood pressure.

**Table 3 jcm-11-03554-t003:** Treatments at baseline.

	IHD or PAD (*n* = 1089; 100%)	IHD and PAD (*n* = 184; 16.9%)	IHD (*n* = 711; 65.3%)	PAD (*n* = 194; 17.8%)	P_IHD_ vs. _PAD_
**Antiplatelet agents**					
Antiplatelet therapy	98.3%	98.9%	98.3%	97.9%	0.759
Single	80.6%	75.8%	80.5%	93.2%	<0.001
Dual	18.2%	24.7%	20.0%	6.8%	<0.001
Single antiplatelet therapy					
Aspirin	87.4%	79.7%	91.3%	80.8%	<0.001
Clopidogrel	10.9%	20.3%	6.7%	16.9%	<0.001
Ticagrelor	1.5%	1.4%	2.0%	0	0.075
Dual antiplatelet therapy					
Aspirin + clopidogrel	58.1%	68.9%	50.7%	100.0%	<0.001
Aspirin + ticagrelor	38.4%	24.4%	46.4%	0	0.001
Aspirin + prasugrel	2.5%	0	3.6%	0	>0.999
Duration of antiplatelet therapy, months (per patient)	56.5 ± 56.1	69.5 ± 65.0	58.7 ± 56.2	37.2 ± 41.2	<0.001
Single	62.3 ± 56.1	80.3 ± 66.6	65.6 ± 55.2	38.8 ± 42.1	<0.001
Dual	32.4 ± 47.4	33.2 ± 43.3	33.4 ± 50.0	21.6 ± 28.4	0.278
**Lipid lowering therapy**					
≥1 lipid lowering drugs	96.9%	97.3%	98.3%	91.2%	<0.001
Statins	94.9%	95.7%	96.1%	89.7%	0.001
Ezetimibe	38.0%	41.8%	43.9%	12.9%	<0.001
Fibrate	4.8%	5.4%	4.8%	4.1%	0.848
PCSK9 inhibitors	2.7%	2.2%	3.2%	1.0%	0.136
Others	0.6%	0.5%	0.6%	0.5%	>0.999
Atorvastatin	61.2%	63.6%	60.0%	63.2%	0.487
Dose, mg	51.9 ± 23.8	56.2 ± 23.8	51.8 ± 23.3	48.3 ± 25.0	0.118
Duration, months	44.4 ± 45.0	51.6 ± 57.0	46.4 ± 44.6	31.9 ± 30.6	0.003
Rosuvastatin	24.6%	22.7%	28.0%	13.2%	<0.001
Dose, mg	17.5 ± 5.7	17.6 ± 4.5	17.5 ± 5.8	16.5 ± 6.3	0.511
Duration, months	30.5 ± 32.2	34.8 ± 36.0	28.8 ± 30.9	34.9 ± 36.0	0.946
**RAAS inhibitors**					
RAAS inhibitors (can be combined with AA)	73.5%	76.6%	75.1%	64.4%	0.004
ACEi	37.6%	32.1%	41.1%	30.4%	0.008
ARB	32.8%	41.8%	30.4%	33.0%	0.485
MRA	5.3%	6.5%	6.0%	1.5%	0.009
ARNI	5.0%	7.1%	5.8%	0	<0.001
**Antianginal drugs**					
≥1 antianginal drugs	79.7%	93.5%	84.4%	49.5%	<0.001
Beta blockers	63.4%	76.6%	73.4%	13.9%	<0.001
Calcium channel blockers	23.0%	29.9%	19.8%	28.4%	0.014
Nitrates	11.7%	22.8%	11.5%	1.5%	<0.001
Ivabradine	5.3%	5.4%	6.6%	0.5%	<0.001
Ranolazine	5.1%	7.6%	5.9%	0	<0.001
Cilostazol	3.3%	5.4%	0	13.4%	<0.001
**Revascularization procedures**					
Coronary revascularization	74.6%	81.0%	93.2%	0	<0.001
Coronary percutaneous intervention	67.2%	67.9%	85.4%	0	<0.001
Bypass surgery	11.5%	19.0%	12.7%	0	<0.001
Revascularization procedure for PAD	12.1%	33.2%	0.1%	36.1%	<0.001
Carotid revascularization	5.6%	12.5%	0.1%	19.1%	<0.001

ARNI: angiotensin receptor and neprilysin inhibition; ACEi: Angiotensin Converting Enzyme Inhibitors; ARB: angiotensin receptor blockers; IHD: ischemic heart disease; MRA: mineralocorticoid receptor antagonists; PAD: peripheral artery disease; RAAS: renin angiotensin system inhibitor.

**Table 4 jcm-11-03554-t004:** Proportion of patients achieving cardiovascular risk factor targets.

	IHD or PAD (*n* = 1089; 100%)	IHD and PAD (*n* = 184; 16.9%)	IHD (*n* = 711; 65.3%)	PAD (*n* = 194; 17.8%)	P_IHD_ vs. _PAD_
BP < 140/90 mmHg	62.6%	62.4%	67.9%	43.0%	<0.001
BP < 130/80 mmHg	29.7%	27.7%	34.1%	15.7%	<0.001
BP < 120/70 mmHg	7.4%	7.8%	7.7%	5.8%	0.559
LDLc < 70 mg/dL	50.9%	50.4%	53.1%	41.5%	0.033
LDLc < 55 mg/dL	25.9%	31.6%	26.5%	16.0%	0.025
LDLc < 50 mg/dL	18.1%	24.1%	17.3%	14.2%	0.476
HbA1c < 7% (with SGLT2i or GLP1-RA)	17.6%	14.5%	21.7%	8.8%	0.032
HbA1c < 6.5% (with SGLT2i or GLP1-RA)	10.5%	6.0%	13.6%	7.0%	0.245

BP: blood pressure; GLP1-RA: Glucagon-like peptide 1 receptor agonists; HbA1c: glycated hemoglobin; LDLc: LDL cholesterol; SGLT2i: Sodium-glucose Cotransporter-2 Inhibitors.

**Table 5 jcm-11-03554-t005:** Adherence to 2013 ESC guidelines recommendations about the use of cardiovascular prevention drugs.

**Low-dose aspirin or clopidogrel**	77.8%
**Statins**	97.7%
**Statins and low-dose aspirin or clopidogrel**	75.6%
**Patients with HF or LV dysfunction treated with ACEi or ARB**	6.8%
**Patients with hypertension or diabetes treated with ACEi or ARB**	66.4%

ACEi: Angiotensin Converting Enzyme Inhibitors; ARB: angiotensin receptor blockers; HF: heart failure; LV: left ventricular.

**Table 6 jcm-11-03554-t006:** Actions recommended/taken by the physicians during the visit.

	IHD or PAD (*n* = 1089; 100%)	IHD and PAD (*n* = 184; 16.9%)	IHD (*n* = 711; 65.3%)	PAD (*n* = 194; 17.8%)	P_IHD_ vs. _PAD_
Nonpharmacologic recommendations					
Quit smoking	56.0%	63.6%	53.0%	59.8%	0.104
Regular physical activity	59.1%	53.8%	62.4%	52.1%	0.010
Healthy diet	71.3%	63.6%	77.4%	56.2%	<0.001
Weight loss	32.2%	31.5%	34.3%	25.3%	0.019
Modification of treatments					
Antihypertensive drugs	69.0%	79.3%	66.2%	69.1%	0.492
Lipid lowering drugs	82.3%	86.4%	82.4%	77.8%	0.146
Antidiabetic drugs * (**)	37.4% (94.5%)	52.7% (89.8%)	31.4% (95.5%)	44.8% (97.8%)	0.001(0.520)
Metformin * (**)	21.9% (58.7%)	27.2% (51.5%)	18.1% (57.8%)	30.9% (69.0%)	<0.001(0.092)
SGLT2i * (**)	11.2% (30.0%)	14.7% (27.8%)	11.5% (36.8%)	6.7% (14.9%)	0.063(<0.001)
GLP1-RA * (**)	2.7% (7.1%)	3.8% (7.2%)	2.3% (7.2%)	3.1% (6.9%)	0.443(>0.999)
Sulphonylureas * (**)	3.3% (8.8%)	4.3% (8.2%)	3.0% (9.4%)	3.6% (8.0%)	0.641(0.827)
DPP4i * (**)	9.6% (25.6%)	16.3% (30.9%)	7.0% (22.4%)	12.4 (27.6%)	0.025(0.374)
Biguanides * (**)	4.8% (12.8%)	7.6% (14.4%)	4.6% (14.8%)	2.6% (5.7%)	0.232(0.033)
Other interventions					
Cardiac rehabilitation	13.1%	10.9%	17.3%	0	<0.001
Flu vaccine	44.2%	50.5%	45.9%	32.0%	0.001
Pneumococcal vaccine	20.8%	27.7%	21.2%	12.4%	0.005

GLP1-RA: Glucagon-like peptide 1 receptor agonists; SGLT2i: Sodium-glucose Cotransporter-2 Inhibitors. The first proportion * was calculated considering the whole population, whereas the second proportion ** was calculated considering only patients with diabetes.

## Data Availability

Data supproting can be obtained under request.

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
