# Peer review of "Clinical Profile and Management of Patient Patients with Ischemic Heart Disease and/or Peripheral Artery Disease in Clinical Practice: The APALUSA Study"

_jcm, 2022, doi:10.3390/jcm11123554_

Round 1

Reviewer 1 Report

The purpose of the manuscript is to assess the clinical profile and management of patients with ischemic heart disease with or without peripheral artery disease. It was a cross-sectional study encapsulating patients from 80 hospitals throughout Spain. Although the authors do not have any control group in this study which made it difficult to bring any conclusive decision to the table, on a brighter note at least their study provides some useful information related to cardiovascular risk factors control. In my opinion, the manuscript is suitable for publication in the “Journal of Clinical Medicine” only after solving the mentioned minor issues.

Minor points:

1.      In the first paragraph of section “method”, authors are requested to provide a table providing all the criteria for patients/conditions for inclusion in the study. That table should replace a major portion of that paragraph.  

2.      Add another such table while discussing how the proportion of patients that achieved cardiovascular risk factors targets was determined. For example: how blood pressure, lipid, diabetes, antiplatelet therapy, stent, drugs, etc. That table should replace a few paragraphs or at least shorten those.

3.      Section 2.1 should be elaborated extensively. All data acquired from statistical analysis needs to be explained along with their implications.

Author Response

The purpose of the manuscript is to assess the clinical profile and management of patients with ischemic heart disease with or without peripheral artery disease. It was a cross-sectional study encapsulating patients from 80 hospitals throughout Spain. Although the authors do not have any control group in this study which made it difficult to bring any conclusive decision to the table, on a brighter note at least their study provides some useful information related to cardiovascular risk factors control. In my opinion, the manuscript is suitable for publication in the “Journal of Clinical Medicine” only after solving the mentioned minor issues.

We agree with the reviewer that the lack of a control group is a limitation of the study. This was stated in the manuscript.

Minor points:

  1. In the first paragraph of section “method”, authors are requested to provide a table providing all the criteria for patients/conditions for inclusion in the study. That table should replace a major portion of that paragraph.  

A table with the inclusion/exclusion criteria has been included (new table 1).

  1. Add another such table while discussing how the proportion of patients that achieved cardiovascular risk factors targets was determined. For example: how blood pressure, lipid, diabetes, antiplatelet therapy, stent, drugs, etc. That table should replace a few paragraphs or at least shorten those.

CV risk factors control targets have been included in table 1.

  1. Section 2.1 should be elaborated extensively. All data acquired from statistical analysis needs to be explained along with their implications.

The statistical analysis section has been expanded in odder to be more clear.

Reviewer 2 Report

This is an observational cross sectional study performed in 80 hospital in Spain on the adherence to treatment guidelines in patients with ischaemic heart disease and peripheral artery disease. Its strength is a relatively large sample size. 

I have the following comments

1.     This cross-sectional study was said to be developed in the departments of cardiology (n=45), internal medicine (n=20) and vascular surgery (n=15) of 80 hospitals throughout Spain. With this large basis of recruitment how is it that only 1112 patients were examined? I could not find the period of recruitment. Please explain.

2.     In spite of this relatively large sample size, the authors made no efforts to identify the determinants of the poor control of risk factors in Spain. This is a major limitation of the study that must be addressed.

3.     Moreover, they failed to discuss the differences in the health care and hospital organization between Spain and the other countries that might contribute to this poor implementation of guidelines into practice. For example, they should present guidelines are disseminated in Spain, how adherence to guidelines is verified, what type of CME is exploited, etc..

4.     The Authors focused on IHD and PAD and ignored atherosclerotic renal artery stenosis, which is a marker of a slim outcome.

5.     Likewise, I would expected a high rate of atrial fibrillation in this cohort, which obviously might have an impact on treatment. This is totally neglected.

6.     Data on homocysteine must be provided.

7.     The manuscript would benefit from a better search of the literature. A number of relevant recent references were overlooked as, for example, the ARIC study.

8.     Likewise, years back a sub-study of the Genica by Cesari et al in JCVP documented the poor control of lipids in patients selected to undergo coronary angiography. This needs to be quoted and the reasons for the improvement over 20 years in this area should be discussed.

9.     How many of the patients with hypertension had renin and aldosterone measured? What was the rate of hypokalemia? How many had primary aldosteronism physiology and might have undetected primary aldosteronism? What was the proportion of those achieving BP control among the latter? 

10.  This key information should be provided as a recent publication from the AVIS-2 database showed a high proportion of resistant hypertension among those with PA. and accumulating evidence from the Framingham and the ARIC studies suggests a role for aldosterone and renin suppression as a predictor of CV events.

11.  This reviewer would like to know how many of the patients without adequate BP control, i.e. values > 130/80 were on antiplatelet.

Additional issues

·      Table 1: decimals for SBP, DBP, cholesterol, and TG are a straightforward example of false precision.

·      According to Table 3, about 70% of the patients would be classified as resistant hypertension according to the AHA criteria. 

·      Moreover, the manuscript is plenty of awkward sentences and needs a major revision of the English that is poor in several parts. For example (page 9): In table 5 are presented the actions recommended/taken during the visit by the physicians. Should be:  Table 5 shows the actions recommended/taken by the physicians during the visit.

·      Table 2. AA is an incorrect term: please replace with MRA.

·      Table 5: again: issues of false precision. Why 2 percentages for the antidiabetics? Figures for biguanides are missing.

Author Response

This is an observational cross sectional study performed in 80 hospital in Spain on the adherence to treatment guidelines in patients with ischaemic heart disease and peripheral artery disease. Its strength is a relatively large sample size. 

I have the following comments

  1. This cross-sectional study was said to be developed in the departments of cardiology (n=45), internal medicine (n=20) and vascular surgery (n=15) of 80 hospitals throughout Spain. With this large basis of recruitment how is it that only 1112 patients were examined? I could not find the period of recruitment. Please explain.

The first patient was recruited on 20th November 2019 and the last patient on 4th February 2021. The recruitment of patients was affected by COVID-19 pandemic.

  1. In spite of this relatively large sample size, the authors made no efforts to identify the determinants of the poor control of risk factors in Spain. This is a major limitation of the study that must be addressed.

The poor risk factors control is not a limitation of the study by itself, but a worrying result that we have strengthened in the manuscript.

  1. Moreover, they failed to discuss the differences in the health care and hospital organization between Spain and the other countries that might contribute to this poor implementation of guidelines into practice. For example, they should present guidelines are disseminated in Spain, how adherence to guidelines is verified, what type of CME is exploited, etc..

In this study, we focused on Spain. This is a limitation for the generalization of the results, particularly to countries with different clinical profile of patients and healthcare systems. We have included this limitation in the study. On the other hand, our data are poor, but in line with other registries, such as EUROASPIRE V and DA VINCI registries.

  1. The Authors focused on IHD and PAD and ignored atherosclerotic renal artery stenosis, which is a marker of a slim outcome.

We agree with the reviewer, and this is a limitation of the study.

  1. Likewise, I would expected a high rate of atrial fibrillation in this cohort, which obviously might have an impact on treatment. This is totally neglected.

Patients with a history of atrial fibrillation or other indication for anticoagulation were excluded from the study, as we wanted to analyze how the antiplatelets drug were prescribed in patients that did not require anticoagulation. 

  1. Data on homocysteine must be provided.

Unfortunately, homocysteine levels were not determined.

  1. The manuscript would benefit from a better search of the literature. A number of relevant recent references were overlooked as, for example, the ARIC study.

      The ARIC study has been now included in the discussion.

  1. Likewise, years back a sub-study of the Genica by Cesari et al in JCVP documented the poor control of lipids in patients selected to undergo coronary angiography. This needs to be quoted and the reasons for the improvement over 20 years in this area should be discussed.

This manuscript has also been included in the discussion.

  1. How many of the patients with hypertension had renin and aldosterone measured? What was the rate of hypokalemia? How many had primary aldosteronism physiology and might have undetected primary aldosteronism? What was the proportion of those achieving BP control among the latter? 

Although this would have been very interesting, these points were note determined, as they were not the objective of the study.

  1. This key information should be provided as a recent publication from the AVIS-2 database showed a high proportion of resistant hypertension among those with PA. and accumulating evidence from the Framingham and the ARIC studies suggests a role for aldosterone and renin suppression as a predictor of CV events.

Unfortunately, this was not determined.

  1. This reviewer would like to know how many of the patients without adequate BP control, i.e. values > 130/80 were on antiplatelet.

 Considering that 98.3% of patients were taking at least one antiplatelet agent, and that only 18 patients were not taking any antiplatelet agents, it is not expect that differences could be found according to blood pressure control.

Additional issues

  • Table 1: decimals for SBP, DBP, cholesterol, and TG are a straightforward example of false precision.

      Numbers have been rounded to make the calculation easier.

  • According to Table 3, about 70% of the patients would be classified as resistant hypertension according to the AHA criteria. 

To make a precise definition of the number of patients with resistant hypertension, the dose of the antihypertensive agents should also have been determined. Anyway, the proportion of patients with this condition seems high.

  • Moreover, the manuscript is plenty of awkward sentences and needs a major revision of the English that is poor in several parts. For example (page 9): In table 5 are presented the actions recommended/taken during the visit by the physicians. Should be:  Table 5 shows the actions recommended/taken by the physicians during the visit.

A native English speaker has reviewed the manusctipt.

  • Table 2. AA is an incorrect term: please replace with MRA.

It has been corrected.

  • Table 5: again: issues of false precision. Why 2 percentages for the antidiabetics? Figures for biguanides are missing.

The first proportion was calculated considering the whole population, whereas the second proportion was calculated considering only patients with diabetes. This as been clarified in the manuscript.

Round 2

Reviewer 2 Report

The authors have addressed some of my criticisms, but failed to discuss the reasons underlying this poor control of the CV risk factors.

1. In particular, they should discuss if this has anything to do with the Spanish organisation of the health care system. For example, drug prescription, reimbursement of drug expenses, poor doctor/patients education, concerns about side effects, etc..

2. The recruitment needs to be better presented. From what one can gather the patients were referred for a follow-up visit after an hospital admission. This should be presented and discussed along with the rate of patients who did not present at this visit. I would imagine that in this cohort of lost to follow-up the rate of CV risk control could have been much worse as is at the level of GPs.

These issues must be better discussed.

Moreover, there are words/sentences that are useless.

Delete 'post-authorisation.

Delete the first para in the Statistical analysis as this is an observational study that does not test a specific hypothesis. As such it does not need power calculation.

Author Response

The authors have addressed some of my criticisms, but failed to discuss the reasons underlying this poor control of the CV risk factors.

  1. In particular, they should discuss if this has anything to do with the Spanish organisation of the health care system. For example, drug prescription, reimbursement of drug expenses, poor doctor/patients education, concerns about side effects, etc..

As there is reimbursement for the great majority of cardiovascular drugs in Spain, the main reasons for poor cardiovascular risk factors control may include insufficient medical education, the wrong perception that patients were adequately controlled that led to an inadequate intensification of treatment and the underestimation of cardiovascular risk.

  1. The recruitment needs to be better presented. From what one can gather the patients were referred for a follow-up visit after an hospital admission. This should be presented and discussed along with the rate of patients who did not present at this visit. I would imagine that in this cohort of lost to follow-up the rate of CV risk control could have been much worse as is at the level of GPs.

This was an observational and cross-sectional study, with only one visit that coincided with any of the follow-up routine visits of the patients according to clinical practice. Patients were consecutively included as patients met with the inclusion/exclusion criteria. Unfortunately, it was not recorded the proportion of patients that did not present at this visit and this could have overestimated the proportion of patients that reached the targets.

Moreover, there are words/sentences that are useless. Delete 'post-authorisation. Delete the first para in the Statistical analysis as this is an observational study that does not test a specific hypothesis. As such it does not need power calculation.

These sentences/words have been deleted.